# CausalDiffusion: A Causality-Embedded Diffusion Model for Cross-Modal Physiological Signal Synthesis

## Abstract

Synthesizing high-fidelity, physiologically plausible signals offers transformative solutions to data scarcity and privacy concerns in biomedical research and clinical care. While Denoising Diffusion Probabilistic Models (DDPM) excel at producing statistically plausible signals, they often fail physiological validation due to neglecting the underlying causal mechanisms governing cardiovascular dynamics. To bridge this gap, we introduce CausalDiffusion, a novel Causality-Embedded DDPM that learns and embeds dynamic causal knowledge directly into the reverse diffusion process. One core of CausalDiffusion is to learn sample-specific causal graphs by dynamically weighting a static base graph, which is constructed offline by synergizing domain knowledge with causal structures discovered by the Fast Causal Inference (FCI) algorithm. These adaptive causal embeddings modulate the U-Net denoiser at multiple scales, transforming causal graphs into structured regularizers that constrain the generative pathways to physiologically plausible manifolds. On the challenging task of continuous blood pressure waveform generation from ECG and PPG inputs, CausalDiffusion achieves state-of-the-art performance, demonstrating remarkable robustness, particularly under arrhythmic conditions where pure data-driven models degrade. Furthermore, we showcase the framework's generality by applying it in the PPG-to-ECG synthesis task, where it achieves superior performance in generating complex waveform morphology. By synergizing the generative power of diffusion models with the rigor of causal reasoning, our work establishes a new paradigm for building reliable and interpretable generative models in biomedicine and beyond. Our data and code are publicly available at `https://github.com/Tasksqwx/CausalDiffusion`.

## 1 Introduction

The synthesis of high-fidelity, physiologically plausible signals — such as Electrocardiogram (ECG), Photoplethysmogram (PPG), and Arterial Blood Pressure (ABP) waveforms — offers a transformative solution to data scarcity and privacy constraints in biomedical research and clinical practice (Neifar et al. (2025)). Access to large-scale, multi-modal physiological signals is essential for developing robust clinical AI, yet it is often hampered by the difficulty of data collection and stringent patient confidentiality regulations. Generative models offer a transformative solution by enabling the creation of realistic, synthetic data surrogates. To this end, a variety of deep generative architectures have been explored for physiological signal synthesis, including autoregressive models like WaveNet(Paviglianiti et al. (2022)), convolutional networks such as U-Net(Maryam et al. (2024)), Generative Adversarial Networks (GANs)(Chen et al. (2022)), and more recent sequence-to-sequence approaches based on Transformers(Lan (2023)) and state-space models like Mamba(Gu & Dao (2023)). Denoising Diffusion Probabilistic Models (DDPM) (Ho et al. (2020)) have emerged as a new state-of-the-art in generative modeling, demonstrating remarkable success in synthesizing high-fidelity data across diverse modalities, particularly in perceptual domains such as image and audio generation (Rombach et al. (2022); Liu et al. (2023)). Recently, DDPM have also been applied to physiological signal synthesis, with promising results in tasks like generating ABP waveforms from ECG and/or PPG inputs(Ma et al. (2024a); Liu et al. (2024b)), as well as reconstructing

ECG signals from PPG data (Alcaraz & Strodthoff (2023); Shome et al. (2024); Adib et al. (2023)). These initial applications have demonstrated the potential of DDPM to tackle challenges in biomedical signal generation. However, despite their effectiveness, DDPM encounter significant limitations when applied to physiological domains. Their purely data-driven paradigm, while excelling in generating visually convincing outputs, often lacks transparency, interpretability, and consistency with known physical or physiological processes, which is critical for physiological signal synthesis. For instance, in generating ABP waveforms from ECG and PPG inputs, a standard DDPM might produce visually convincing waveforms but include physiologically implausible artifacts. One typical artifact is generating a dicrotic notch preceding the systolic peak, a clear violation of cardiovascular dynamics(O'Rourke (1995)). These inconsistencies highlight the critical need for generative models that go beyond statistical fidelity and incorporate domain-specific causal knowledge to ensure physiological plausibility.

Causal-Informed Deep Learning and Physics-Informed Neural Networks (PINNs) have emerged as a promising paradigm to incorporate domain knowledge into neural architectures (Yang et al. (2021); Raissi et al. (2019)). Recent advancements, such as CiGNN (Liu et al. (2024a)), demonstrate how causal graphs can enhance physiological signal processing, while causal representation learning approaches (Bengio et al. (2020); Schölkopf et al. (2021)) highlight the advantages of leveraging causal structures in learned representations. Additionally, the Physics-Informed diffusion model has also been proposed to generate complex physical phenomena, such as blood flow fields (Qiu et al. (2024)), showcasing the potential of integrating physical knowledge into generative frameworks. Despite these advances, existing methods for incorporating prior knowledge into deep learning models, such as PINNs (Raissi et al. (2019)), often rely on global constraints or simplified conditioning schemes. For example, PINNs typically embed differential equations as soft constraints within the loss function, enforcing physical consistency indirectly. While effective in certain scenarios, these approaches lack the granularity needed to explicitly model the multi-dimensional causal relationships that govern complex physiological systems. This limitation is particularly critical in physiological signal synthesis, where the interplay between different modalities (e.g., ECG, PPG, and ABP) is governed by highly dynamic and nonlinear causal mechanisms.

To address this gap, we introduce a framework that directly embeds sample-specific causal knowledge into the generative process. By explicitly modeling the causal relationships that capture intrinsic physiological dynamics, our framework produces more robust and physiologically plausible signals. To this end, we introduce the Causality-Embedded Denoising Diffusion Probabilistic Model (CausalDiffusion), which pioneers the integration of explicit and dynamic causal graphs directly into diffusion models for complex, cross-modal physiological time-series generation. At its core is the construction of a dynamic causal graph for each sample, which learns a representation of the underlying causal system to produce adaptive embeddings that actively modulate the U-Net denoiser at multiple feature scales. These embeddings act as learned, structured regularizers that dynamically constrain generative paths to physiologically valid manifolds.

Our main contributions are fourfold: (1) We propose CausalDiffusion, a novel framework that integrates dynamic, end-to-end learned causal graphs into diffusion models' denoising processes. This approach adaptively enforces domain-specific constraints during the denoising process. (2) We design a novel causal guidance mechanism composed of two key innovations: an offline Hybrid Enhancement framework that constructs a robust base causal graph by fusing domain knowledge with data-driven insights from Fast Causal Inference (FCI)(Glymour et al. (2019)), and an online Graph Convolutional Network (GCN)(Kipf & Welling (2017)) that transforms this static prior into sample-specific guidance. (3) Through extensive experiments on both public and self-collected datasets, we demonstrate that CausalDiffusion achieves state-of-the-art performance in high-fidelity ABP waveform generation from ECG and PPG inputs, exhibiting exceptional robustness under challenging conditions, such as arrhythmias. (4) We showcase the generality of our framework by successfully applying it to a distinct cross-modal task of synthesizing ECG from PPG input, highlighting its potential as a general-purpose tool for causality-embedded generative modeling.

## 2 METHODOLOGY

### 2.1 PROBLEM FORMULATION

We address the cross-modal physiological signal synthesis as learning a generative mapping from input physiological signals to the target signal within the same biological system. Unlike simple signal translation, this task must preserve the underlying physiological relationships and temporal dynamics that govern multi-modal biological processes. Formally, given a set of input signals $X = \{x_1, \ldots, x_C\} \in R^{C \times L}$, where $C$ is the number of channels, and $L$ is the signal length, our goal is to learn a generative model capable of synthesizing a target signal $z \in R^L$ that is physiologically consistent with the inputs.

We reformulate this task as a conditional generation problem. Instead of learning a direct mapping, the conditional generative model can be expressed as follows: $p_\theta(z|X, G)$, where $G = (V, E)$ is a causal graph that explicitly encodes the physiological mechanisms linking input and target signals, $\theta$ is the model parameter. This graph, with nodes $V$ representing physiological features and edges $E$ representing causal influences, acts as a strong sample-specific prior that guides the generation process and ensures synthesized waveforms conforming to known physiological constraints.

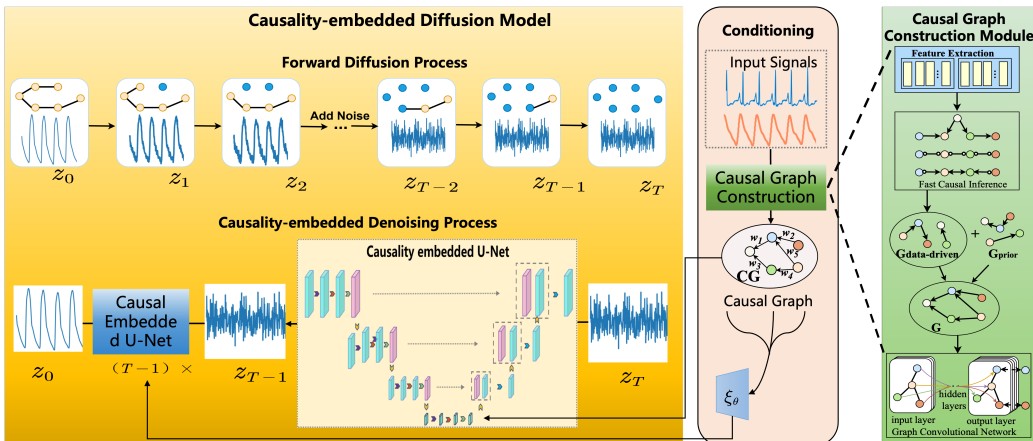

Figure 1: The overall architecture of CausalDiffusion. The framework comprises a forward diffusion process and a causality-embedded reverse denoising process.

### 2.2 CAUSALITY-EMBEDDED DIFFUSION FRAMEWORK

The overall framework of the proposed CausalDiffusion is presented in Figure 1, which comprises a forward diffusion process and a causality-embedded reverse denoising process. The forward diffusion process $q(z_t|z_{t-1})$ can be characterized as a Markov chain that progressively adds Gaussian noise at every timestep $t$:

$$q(z_T|z_0) := \prod_{t=1}^{T} q(z_t|z_{t-1}), \quad where \ q(z_t|z_{t-1}) := \mathcal{N}(z_t; \sqrt{1 - \beta}z_{t-1}, \beta_t I). \quad (1)$$

Here, $z_0 \sim q(z_0)$ is a clean signal and $\beta_t$ is a small positive constant obtained from a fixed variance schedule $\beta_1, \cdots, \beta_T$. Let $\alpha_t := 1 - \beta_t$ and $\bar{\alpha}_t := \prod_{s=1}^{t} \alpha_s$. Then, the forward diffusion process yields a sample at timestep $t$, denoted by $z_t$, which can be obtained in a single step as follows:

$$z_t = \sqrt{\bar{\alpha}_t}z_0 + \sqrt{1 - \bar{\alpha}_t}\epsilon, \quad where \ \epsilon \sim \mathcal{N}(0, I). \quad (2)$$

Since the reverse step of the forward process, $q(z_{t-1}|z_t)$ is computationally infeasible, DDPM maximizes the variational lower bound (ELBO) using a parameterized Gaussian transition $p_\theta(z_{t-1}|z_t)$

with parameter $\theta$. Consequently, the reverse process is approximated as a Markov chain with a learned mean and a fixed variance, initiated from $p(z_T) = \mathcal{N}(z_T; 0, I)$:

$$p_\theta(z_{0:T}) := p_\theta(z_T) \prod_{t=1}^{T} p_\theta(z_{t-1}|z_t), \tag{3}$$

Where

$$p_\theta(z_{t-1}|z_t) := \mathcal{N}(z_{t-1}; \mu_\theta(z_t, t), \sigma_t^2 I), \quad \mu_\theta(z_t, t) := \frac{1}{\sqrt{\alpha_t}}(z_t - \frac{1 - \alpha_t}{\sqrt{1 - \bar{\alpha}_t}}\epsilon_\theta(z_t, t)) \tag{4}$$

The diffusion model $\epsilon_\theta$ can be conditioned upon the input $X$, which makes the conditional objective:

$$L(\theta) := \mathbb{E}_{t, z_0, \epsilon}[\|\epsilon - \epsilon_\theta(\sqrt{\bar{\alpha}_t}z_0) + \sqrt{1 - \bar{\alpha}_t}\epsilon, X, t)\|^2]. \tag{5}$$

The physiological signal synthesis task can be performed by adopting Equation 5 to generate the corresponding target signal $z_0$ conditional upon the input signals $X$. To enable the Diffusion model to learn both global structures and fine-grained local details of target signals for nuanced bio-signal translation, we introduce a novel framework that embeds structured causal knowledge directly into the denoising process of the Diffusion Model.

**Causality-Emedded Denoising Reverse Process.** Our approach redefines the standard conditional reverse process. We augment the denoising network, $\epsilon_\theta$, to be conditioned not only on the diffusion timestep $t$ and multi-modal signals $X$, but also on a learned causal graph, $G$.

This is achieved by parameterizing the mean of the reverse transition probability, $\mu_\theta$, as a function of the noise prediction network $\epsilon_\theta$. The reverse process is thus defined as:

$$p_\theta(z_{t-1}|z_t, X, G) := \mathcal{N}(z_{t-1}; \mu_\theta(z_t, t, X, G), \sigma_t^2 I) \tag{6}$$

where the mean $\mu_\theta$ is computed using our causally conditioned noise predictor:

$$\mu_\theta(z_t, t, x, G) := \frac{1}{\sqrt{\alpha_t}}\left(z_t - \frac{1 - \alpha_t}{\sqrt{1 - \bar{\alpha}_t}}\epsilon_\theta(z_t, t, x, G)\right) \tag{7}$$

Here, the embedding $G$ is learned by a Causal Graph Construction Module. It encapsulates learned physiological relationships, thereby steering the denoising trajectory towards a physiologically plausible synthesis. The network $\epsilon_\theta$ is trained to predict the noise component $\epsilon$ from the noisy signal $z_t$, given all conditioning inputs.

**Multi-Objective Physiological Regularization.** Our model is trained end-to-end with a multi-objective loss function designed to enforce both statistical fidelity and physiological plausibility. The total loss $\mathcal{L}$ is a weighted sum of the diffusion loss and an auxiliary physiological loss:

$$\mathcal{L} = \mathcal{L}_{diff} + \lambda_{aux}\mathcal{L}_{aux} \tag{8}$$

where the weighting coefficient $\lambda_{aux}$ is empirically set to 0.1 in our study.

The diffusion loss $\mathcal{L}_{diff}$ combines L1 and L2 norms to accurately predict the added noise $\epsilon$. This hybrid loss leverages the L1 norm's robustness to outliers and the L2 norm's promotion of smoothness:

$$\mathcal{L}_{diff} = \mathbb{E}_{t, z_0, \epsilon}\left[\|\epsilon - \epsilon_\theta(z_t, t, X, G)\|_2^2 + \lambda_{L1}\|\epsilon - \epsilon_\theta(z_t, t, X, G)\|_1\right] \tag{9}$$

The auxiliary loss, $\mathcal{L}_{aux}$, serves to construct and refine a physiologically meaningful causal graph embedding, $G$. This process is supervised by the ground-truth physiological components. First, let $d$ be the vector of physiological components extracted from the ground-truth signal $z_0$. We then employ a Component Predictor network, $\mathcal{P}$, which learns to predict these components directly from the input conditioning signals $x$. The predicted component vector is thus $\bar{d} = \mathcal{P}(x)$. The auxiliary loss is defined as the Mean Squared Error (MSE) between the true and predicted components, directly optimizing the Component Predictor:

$$\mathcal{L}_{aux} = \mathbb{E}_{z_0, x}\left[\|d - \mathcal{P}(x)\|_2^2\right] \tag{10}$$

This joint optimization ensures that the causal graph embedding $G$, which is constructed using the predicted components $\bar{d}$, is grounded in physiologically accurate estimations. This, in turn, provides a more effective and interpretable structural prior to guide the primary diffusion task.

## 2.3 Causal Graph Construction Module

The cornerstone of our CausalDiffusion framework is a causal graph, represented as a directed graph $G = (V, E)$. This graph encodes the assumed direct causal relationships among a set of $N$ physiological variables $V = \{v_1, \ldots, v_N\}$. To construct this graph, we introduce a Hybrid Enhancement framework that synergies domain knowledge with data-driven knowledge, overcoming the inherent limitations of relying on either source alone.

We first establish a prior causal graph, $G_{prior} = (V, E_{prior})$, derived from established physiological principles. The node set $V$ comprises features extracted from the input signals and clinical key components of the target signal. A directed edge $(v_i, v_j) \in E_{prior}$ exists if and only if there is a strong, pre-existing scientific consensus positing a direct causal influence from variable $v_i$ to $v_j$. This graph is represented by its adjacency matrix $A_{prior} \in \{0, 1\}^{N \times N}$. $G_{prior}$ serves as a reliable but potentially incomplete causal skeleton. The full list of selected features and their corresponding target components, which form the basis of our a prior causal skeleton, is detailed in the Appendix (Appendix Table 3, Appendix Table 4, Appendix Table 5).

To augment the prior with insights specific to our dataset, we learn a data-driven causal graph, $G_{data-driven}$, directly from the training data. Our first step is to represent the entire training set as a single data matrix $D$. This matrix has dimensions $M \times N$, where $M$ is the total number of training samples, and $N$ is the number of physiological variables (or features) to be analyzed. Each row of $D$ corresponds to one sample, and each column corresponds to one variable from our set $V$. With this data matrix, our goal is to uncover the underlying causal relationships between the variables. We employ the Fast Causal Inference (FCI) algorithm(Glymour et al. (2019)), a robust constraint-based method that handles unobserved confounders by performing a series of conditional independence (CI) tests. A CI test assesses the null hypothesis $H_0 : v_i \perp\!\!\!\perp v_j \mid S$ for a given conditioning set $S \subseteq V \setminus \{v_i, v_j\}$. For continuous variables, this test is often the Fisher's z-transformation of the partial correlation coefficient, $\rho_{ij \cdot S}$. The test statistic is given by:

$$z(v_i, v_j \mid S) = \frac{1}{2} \sqrt{M - |S| - 3} \cdot \ln \left( \frac{1 + \rho_{ij \cdot S}}{1 - \rho_{ij \cdot S}} \right) \tag{11}$$

where $|S|$ is the size of the conditioning set. The null hypothesis $H_0$ is rejected if $|z| > \Phi^{-1}(1 - \alpha/2)$, where $\Phi^{-1}$ is the inverse of the standard normal cumulative distribution function and $\alpha$ is the significance level. To ensure the invertibility of the covariance matrix required to compute $\rho_{ij \cdot S}$, we first mitigate multicollinearity by recursively eliminating one variable from any pair $(v_i, v_j)$ with a Pearson correlation $|\rho_{ij}| > \tau$. The FCI algorithm is then applied to the reduced set of variables. The output is a Partially Ancestral Graph (PAG), from which we extract the set of discovered directed edges, $E_{data}$, and its corresponding adjacency matrix, $A_{data}$.

The final step fuses the prior knowledge with the data-driven knowledge. We construct the ultimate causal graph, $G = (V, E)$, using a knowledge-guided fusion strategy. This process can be formalized as follows: we define the final edge set $E$ as the union of the a prior edges and the data-driven edges.

$$E = E_{prior} \cup E_{data-driven} \tag{12}$$

This fusion operation, defined by the edge set union $E_{base} = E_{prior} \cup E_{data}$, ensures that all trusted prior knowledge is preserved while integrating novel, data-driven relationships. The corresponding ultimate adjacency matrix, $A_{base} \in \{0, 1\}^{N \times N}$, is computed as the element-wise logical OR of the constituent matrices. This static matrix $A_{base}$ serves as a foundational structural skeleton.

To transform this static skeleton into a dynamic, sample-specific guide, we introduce a Graph Modulation Network. For each input sample, a set of initial node features $\{h_i^{(0)}\}_{i=1}^N$ is derived from its physiological characteristics. A global context vector, computed via pooling these features, is then projected by a multilayer perceptron (MLP) to generate a dynamic modulation matrix $M \in \mathbb{R}^{N \times N}$. The final, sample-specific weighted adjacency matrix, $A_{dynamic}$, is obtained by applying this modulation to the static base:

$$A_{dynamic} = A_{base} \odot \sigma(M) \tag{13}$$

where $\odot$ denotes the Hadamard product and $\sigma$ is a scaling activation function. This dynamic graph is then processed by a Graph Convolutional Network (GCN) (Kipf & Welling, 2017), which incorporates the learned edge weights directly into its propagation rule to perform a weighted aggregation

of information:

$$h_i^{(l+1)} = \sigma' \left( \sum_{j \in \mathcal{N}(i) \cup \{i\}} (A_{dynamic})_{ij} \cdot W^{(l)} h_j^{(l)} \right) \tag{14}$$

where $(A_{dynamic})_{ij}$ is the dynamic weight and $W^{(l)}$ is a trainable weight matrix. By stacking multiple such layers, we generate a final set of node embeddings, which are then aggregated to form the final causal vector $G$ that guides the U-Net denoiser.

## 3 Experiments

### 3.1 Experiment Setup

**Datasets.** Following, we present the details of the datasets that are used in this study.

- MIMIC-III Public Dataset: This dataset, derived from a large-scale intensive care unit database (Johnson et al. (2016)), contains multi-modal physiological signals from critically ill patients. The evaluation utilizes 84,110 segments of 10-second duration, comprising synchronized PPG, ECG, and ABP waveforms sampled at 125 Hz. These segments represent standard physiological conditions, providing a robust foundation for model training and evaluation under normal cardiac rhythms (Wang et al. (2023)).

- Proprietary Arrhythmia Dataset: This dataset comprises 36,537 cardiac cycles collected from 48 clinical patients undergoing radiofrequency ablation treatment for cardiac arrhythmias, with synchronized ECG, PPG, and ABP signals sampled at 250 Hz. The dataset contains various arrhythmic patterns, including atrial fibrillation, ventricular ectopy, and other irregular rhythms that challenge conventional signal processing approaches. For consistency across variable cycle lengths, short signal segments undergo zero-padding to a fixed length of 500 sampling points (Liu et al. (2022)).

**Evaluation Metrics.** To provide a comprehensive assessment across different tasks, we evaluate performance using two main categories of metrics. **Waveform Fidelity Metrics.** These metrics quantify the morphological similarity between the generated and ground-truth waveforms. Key examples include Root Mean Square Error (RMSE), Mean Absolute Error (MAE), and correlation coefficient, which capture both global shape fidelity, local detail accuracy, and the consistency with the ground truth. These metrics are applied across all signal generation tasks. **Clinical Parameter Accuracy.** These metrics assess the clinical utility of the generated signals by evaluating the accuracy of clinical key physiological parameters extracted from them. The specific parameters are task-dependent. For the ABP synthesis task, this includes Mean Error (ME) and Standard Deviation of Error (SDE) for Systolic (SBP) and Diastolic (DBP) Blood Pressure, evaluated against the AAMI standard (Stergiou et al. (2018)). For the ECG synthesis task, this encompasses key Heart Rate Variability (HRV) metrics including heart rate (HR) and RMSSD MAE, as well as functional QRS detection metrics like Sensitivity and Positive Predictive Value(PPV).

**Implementation Details.** The training, validation, and test sets were created by splitting the patient cohorts in an 8:1:1 ratio, ensuring that all data from a single patient belongs exclusively to one set to prevent data leakage. All models were implemented using PyTorch and trained on NVIDIA RTX 3090 GPUs. We used the AdamW optimizer with an initial learning rate of $1 \times 10^{-4}$, weight decay of $1 \times 10^{-5}$, and a batch size of 256. Training proceeded for up to 300 epochs with an early stopping mechanism based on validation loss to prevent overfitting. The diffusion process utilized $T = 300$ steps with a cosine noise schedule, which we found to be empirically superior for physiological signals.

## 4 Results

### 4.1 ABP Waveform Generation for ECG and PPG Inputs

We compare CausalDiffusion against multiple state-of-the-art baseline models spanning different architectural paradigms, including traditional deep learning approaches (BiLSTM(Delrio et al.

(2023)), CNN(Kokkhunthod et al. (2024)), WaveNet(Paviglianiti et al. (2022))), specialized biomedical networks (UNet(Maryam et al. (2024); Tian et al. (2024))), and modern sequence architectures (RDAE(Qin et al. (2021)),KD-INformer(Ma et al. (2022)),Transformer(Ma et al. (2024b)), Mamba(Gu & Dao (2023))). The results, presented in Table 1, demonstrate the consistent superiority of our causality embedded approach across both datasets and all evaluation metrics.

Table 1: Performance Comparison on MIMIC-III and Arrhythmia Dataset

| Method | Morphological Metrics | | | Clinical Parameter Accuracy | |
|---|---|---|---|---|---|
| | RMSE($\downarrow$) (mmHg) | MAE($\downarrow$) (mmHg) | Pearson Corr. ($\uparrow$) | SBP(ME $\pm$ SDE)($\downarrow$) (mmHg) | DBP(ME $\pm$ SDE)($\downarrow$) (mmHg) |
| **MIMIC-III Dataset** | | | | | |
| BiLSTM | 11.49 | 8.83 | 0.892 | -2.49 $\pm$ 13.84 | -2.01 $\pm$ 6.82 |
| CNN | 11.47 | 9.21 | 0.902 | -5.92 $\pm$ 15.36 | -2.05 $\pm$ 7.88 |
| UNet | 5.33 | 3.68 | 0.971 | -0.65 $\pm$ 7.80 | -0.70 $\pm$ 3.71 |
| GSW-UNet | 8.83 | 6.51 | 0.928 | -0.55 $\pm$ 10.26 | 0.14 $\pm$ 5.17 |
| WaveNet | 10.46 | 7.85 | 0.897 | 1.64 $\pm$ 11.68 | 0.61 $\pm$ 5.91 |
| Transformer | 7.21 | 4.88 | 0.949 | 0.90 $\pm$ 8.39 | 0.33 $\pm$ 3.93 |
| Mamba | 5.16 | 13.64 | 0.975 | -0.46 $\pm$ 7.30 | -0.07 $\pm$ 3.51 |
| RDAE | - | - | - | 1.65 $\pm$6.64 | -1.28 $\pm$3.74 |
| KD-Informer | - | - | - | 0.03 $\pm$6.38 | 0.02 $\pm$4.49 |
| DDPM | 3.60 | 3.01 | 0.992 | 1.46$\pm$5.49 | 0.01$\pm$3.15 |
| **CausalDiffusion** | **3.36** | **2.77** | **0.993** | **0.90 $\pm$ 5.06** | **-0.80 $\pm$ 2.90** |
| **Arrhythmia Dataset** | | | | | |
| BiLSTM | 7.14 | 2.96 | 0.823 | -4.22 $\pm$ 16.92 | -4.04 $\pm$ 9.13 |
| CNN | 4.89 | 1.60 | 0.914 | -0.77 $\pm$ 8.41 | -4.03 $\pm$ 6.82 |
| UNet | 5.23 | 2.01 | 0.897 | 6.65 $\pm$ 6.57 | 0.30 $\pm$ 6.22 |
| WaveNet | 5.73 | 2.09 | 0.872 | 3.16 $\pm$ 10.92 | -5.01 $\pm$ 7.12 |
| Transformer | 4.38 | 1.19 | 0.943 | 1.32 $\pm$ 6.35 | -3.10 $\pm$ 6.23 |
| Mamba | 4.17 | 1.02 | 0.954 | 0.04 $\pm$ 5.03 | -0.05 $\pm$ 3.20 |
| DDPM | 3.32 | 1.74 | 0.967 | 0.72$\pm$5.77 | -1.25$\pm$3.36 |
| **CausalDiffusion** | **2.74** | **1.21** | **0.985** | **0.20 $\pm$ 4.78** | **-0.16 $\pm$ 2.77** |

**Performance on Standard Conditions (MIMIC-III).** On the MIMIC-III dataset, CausalDiffusion achieves state-of-the-art performance, recording the lowest waveform reconstruction errors (RMSE: 3.36 mmHg, MAE: 2.77 mmHg). This represents a substantial 36.6% improvement over U-Net baseline and 6.67% RMSE improvement over the pure data-driven DDPM. More importantly, our model's BP parameter estimations comfortably satisfy the stringent AAMI standard (SBP: ME = 0.90 $\pm$ 5.06 mmHg, DBP: ME = -0.80 $\pm$ 2.90 mmHg), with both SDE values well below the 8 mmHg threshold. The high degree of correlation and minimal bias in these estimations are further visually confirmed by the scatter and Bland-Altman plots, along with representative waveform fitting examples, provided in the Appendix (Figure 3 and Figure 4).

**Robustness Under Pathological Conditions (Arrhythmia).** The true strength of our causality-embedded approach becomes evident on the challenging Arrhythmia dataset, where irregular cardiac rhythms disrupt the statistical patterns that conventional models rely upon. As shown in Table 1, while most baselines experience significant performance degradation, CausalDiffusion demonstrates remarkable robustness. Our model again achieves the best waveform reconstruction accuracy (RMSE: 2.74 mmHg, 17.47% RMSE improvement over the pure data-driven DDPM) and maintains high stable BP estimations (SBP: ME = 0.20 $\pm$ 4.78 mmHg, DBP: ME = -0.16 $\pm$ 2.77 mmHg), significantly outperforming pure data-driven DDPM. Qualitative examples, provided in the Appendix (Appendix Figure 5), illustrate our model's ability to generate physiologically plausible waveforms that closely track the ground truth, even in the presence of irregular rhythms.

**Ablation Study: The Impact of Causal Guidance.** To isolate and quantify the specific contribution of our core innovation, we conduct an ablation study by directly comparing CausalDiffusion against the standard conditional DDPM baseline in Table 1. This comparison provides compelling evidence for the critical role of causal guidance. The performance degradation upon removing the causal module is substantial and consistent across both datasets. Notably, on the more challenging Arrhythmia dataset, its removal leads to a 21.2% increase in waveform RMSE (from 2.74 to 3.32 mmHg) and a 43.8% increase in MAE (from 1.21 to 1.74 mmHg). BP estimation accuracy

also deteriorates, with the SBP SDE increasing from 4.78 to 5.77 mmHg. This pronounced improvement on the Arrhythmia dataset supports our central hypothesis: causal guidance provides a robust framework for handling out-of-distribution scenarios by constraining the generation process to physiologically plausible manifolds. The causal graph functions as a learned, dynamic regularizer, preventing the model from generating morphologically implausible waveforms, particularly when confronted with the complex temporal patterns characteristic of cardiac arrhythmia.

## 4.2 GENERALITY OF CAUSALDIFFUSION: APPLICATION TO ECG SYNTHESIS FROM PPG

To validate the generality of our framework, we apply it to a distinct cross-modal synthesis task: generating ECG signals from PPG inputs using MIMIC-III dataset. This task demands high fidelity in both waveform morphology and the accurate modeling of heart rate dynamics. We compare our CausalDiffusion against a standard conditional DDPM baseline across a range of diffusion timesteps ($T$). As demonstrated in Table 2, CausalDiffusion consistently and significantly outperforms the standard DDPM, particularly when a sufficient number of denoising steps ($T = 300$) are employed, where it achieves superior performance across all evaluated metrics.

Table 2: Performance Comparison on the PPG-to-ECG Synthesis Task. The best result for each metric is highlighted in bold. The arrows ($\downarrow$, $\uparrow$) indicate whether a lower or higher value is better for that metric.

| Method | Morphological Metrics | | HRV Metrics | | QRS Detection | |
|---|---|---|---|---|---|---|
| | RMSE ($\downarrow$) | Pearson Corr. ($\uparrow$) | HR MAE (bpm) ($\downarrow$) | RMSSD MAE (ms) ($\downarrow$) | Sensitivity ($\uparrow$) | PPV ($\uparrow$) |
| DDPM(T=50) | 0.3700 | 0.8781 | 23.29 | 68.90 | 0.922 | 0.808 |
| DDPM(T=100) | 0.3677 | 0.8832 | 21.25 | 61.11 | 0.925 | 0.827 |
| DDPM(T=200) | 0.3371 | 0.8995 | 12.13 | 38.25 | 0.926 | 0.890 |
| DDPM(T=300) | 0.3202 | 0.9015 | 10.60 | 36.12 | 0.930 | 0.899 |
| CausalDiffusion(T=50) | 0.3403 | 0.8936 | 23.28 | 64.85 | 0.932 | 0.822 |
| CausalDiffusion(T=100) | 0.3305 | 0.8953 | 14.58 | 44.65 | 0.934 | 0.873 |
| CausalDiffusion(T=200) | 0.3240 | 0.8999 | 11.91 | 37.36 | 0.933 | 0.890 |
| **CausalDiffusion(T=300)** | **0.3150** | **0.9064** | **10.13** | **35.41** | **0.939** | **0.905** |

**Quantitative and Qualitative Analysis.** In terms of waveform reconstruction, CausalDiffusion at $T = 300$ achieves a lower RMSE of 0.3150 (vs. DDPM's 0.3202) and a higher Pearson correlation of 0.9064 (vs. 0.9015), indicating superior morphological fidelity. Such a high degree of morphological accuracy is visually confirmed in the Appendix (Figure 7), where our model's generated waveforms closely track the ground truth across six diverse samples. The model's advantage extends to HRV metrics, where it obtains a lower HR MAE of 10.13 bpm and RMSSD MAE of 35.41 ms, suggesting that the structured causal prior helps infer subtle inter-beat variations more accurately. The excellent agreement in heart rate dynamics is further corroborated by a Bland-Altman analysis, provided in the Appendix (Figure 6), which shows a minimal systematic bias of only 0.33 ms. Finally, the improved physiological understanding is reflected in the state-of-the-art QRS detection scores, with a Sensitivity of 0.939 and a PPV of 0.905.

**Influence of Denoising Timesteps.** We observe that the CausalDiffusion consistently outperformed than pure data-driven DDPM as the number of timesteps $T$ increases, demonstrating the robustness of the framework. In summary, the application to PPG-to-ECG synthesis robustly showcases the generality and effectiveness of our causality-embedded framework. CausalDiffusion demonstrates superior capability across morphological, physiological, and heart-rate-related metrics, establishing a new state-of-the-art for this challenging task.

## 5 DISCUSSION

### 5.1 ANALYSIS OF SYNERGISTIC DESIGN AND ROBUSTNESS

The superior performance of CausalDiffusion stems from the synergistic interplay between its generative backbone and its causality-embedded conditioning. As hypothesized in our introduction, purely data-driven models often sacrifice physiological plausibility for statistical fidelity. Figure 2 provides a stark visual confirmation: a standard conditional DDPM generates a waveform with a physiologically impossible artifact—a dicrotic notch preceding the systolic peak. This structural

error, a direct violation of cardiovascular dynamics ((O'Rourke, 1995)), highlights the failure of pure pattern matching, a limitation that is particularly exposed when modeling complex, out-of-distribution signals like those in our Arrhythmia dataset. Our CausalDiffusion framework directly mitigates this failure mode. The hybrid causal graph, which synergizes domain knowledge with data-driven evidence, functions as a powerful structural regularizer. By transforming this static prior into a dynamic, sample-specific guide via a graph modulation network, our model injects causal constraints directly into the denoising process. This constrains the generative pathways to a physiologically plausible manifold, ensuring the correct temporal ordering of morphological features, such as the systolic peak and the dicrotic notch. It is this deep integration of adaptive causal guidance that grants the robustness of the model, particularly in challenging arrhythmia scenarios where reliance on stable, invariant causal relationships is critical.

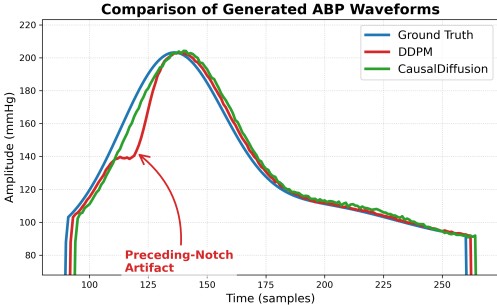 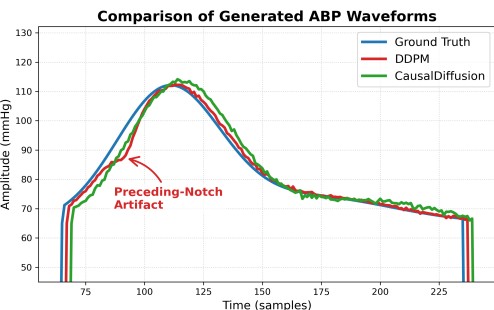

Figure 2: Comparison of generated ABP waveforms against the ground truth. To visually distinguish the highly overlapping curves, the generated waveforms from the Standard DDPM and our CausalDiffusion models have been horizontally shifted by 2 and 4 samples, respectively, for clarity. The plot demonstrates the morphological artifact produced by the standard DDPM (red), while our CausalDiffusion (green) closely tracks the ground truth (blue).

## 5.2 LIMITATIONS AND FUTURE DIRECTIONS

Despite strong results, our approach has several limitations. First, the current causal graph relies on handcrafted features, which may miss informative structure. End-to-end representation learning —for instance, learning feature extractors jointly with causal encoders— could reduce feature bias and improve coverage. Second, our base graph incorporates domain knowledge and statistical associations rather than interventional evidence. Integrating modern causal discovery methods (e.g., differentiable structure learning, time-series causal discovery with latent confounders, or invariant causal prediction) may further improve data-driven graph induction. Finally, the generalizability of our model should also be further strengthened across diverse populations and a broader range of physiological synthesis tasks to enhance reliability and translational value in biomedical research.

## 6 CONCLUSION

In this work, we introduced CausalDiffusion, a novel framework that pioneers the integration of dynamic, end-to-end learned causal knowledge into diffusion models for physiological signal generation. By embedding a sample-specific causal graph directly into the reverse denoising process, CausalDiffusion unites the generative strength of diffusion models with physiologically grounded constraints. Extensive experiments demonstrate that CausalDiffusion achieves state-of-the-art accuracy in continuous ABP waveform generation, particularly under challenging arrhythmic conditions. The generality was also validated in a distinct cross-modal task, enforcing complex ECG morphology from PPG input. This work establishes a promising research direction that combines the generative capabilities of diffusion models with the robustness of causal reasoning. It paves the way for more reliable and interpretable generative models in scientific and clinical applications.

AUTHOR CONTRIBUTIONS

ACKNOWLEDGMENTS

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

## A  APPENDIX: BLOOD PRESSURE WAVEFORM SYNTHESIS DETAILS

This appendix provides supplementary details for the ABP waveform synthesis task, including the feature engineering process, the construction of the prior causal graph, and additional visualizations of the model's performance.

### A.1  FEATURE ENGINEERING AND PRIOR CAUSAL GRAPH CONSTRUCTION

For the ABP waveform synthesis task, the input signals are synchronous PPG and ECG. We initially extract 20 physiologically relevant features following established methodologies (Ding et al. (2019)). The goal is to model four key clinical components of the target BP waveform:

- **Systolic Blood Pressure (SBP)**: Peak arterial pressure during cardiac contraction.
- **Diastolic Blood Pressure (DBP)**: Minimum arterial pressure during cardiac relaxation.
- **Mean Blood Pressure (MBP)**: Average arterial pressure over the cardiac cycle.
- **Pulse Pressure (PP)**: The difference between systolic and diastolic pressures.

To construct the prior causal graph($G_{prior}$) for our model, we performed a correlation analysis to identify the top five most influential features for each of these four components. This process resulted in 13 unique features after deduplication, which form the initial nodes and edges of our causal diagram. The selected features are presented in Table 3. These features encompass pulse width measurements (PW50, PW60), amplitude ratios from second derivative analysis (e.g., $(c + d - b)/a$), and pulse transit times (PTT), capturing a rich set of characteristics relevant to blood pressure estimation.

Table 3: The Five Best Features for Each BP Component, used to construct the base causal graph.

|     | Top Feature 1 | Top Feature 2 | Top Feature 3 | Top Feature 4 | Top Feature 5 |
|-----|---------------|---------------|---------------|---------------|---------------|
| SBP | PW50 | PW_dPPGmax | PW60 | AM_valley_sdPPGd | $(c + d - b)/a$ |
| DBP | AM_dPPGmax_sdPPGb | I_sdPPGb | $(c + d - b)/a$ | $(b - c - d)/a$ | AM_valley_sdPPGb |
| MBP | PW50 | PW_dPPGmax | $(b - c - d)/a$ | $(c + d - b)/a$ | AM_sdPPGb_sdPPGd |
| PP | PTT_R_sdPPGa | PTT_R_dPPGpeak | TD_dPPGpeak_sdPPGc | PW50 | PW_dPPGmax |

### A.2  SUPPLEMENTARY VISUALIZATIONS

To supplement the results presented in the main paper, this section provides additional visualizations of CausalDiffusion's performance on the BP synthesis task. Figure 3 displays the correlation and Bland-Altman plots for SBP and DBP estimations, while Figure 4 and Figure 5 provides qualitative examples of the generated waveforms against the ground truth.

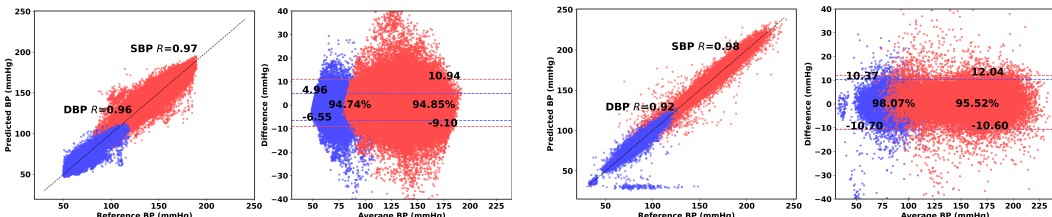

Figure 3: Correlation and Bland-Altman plots of CausalDiffusion for BP estimations on (left) MIMIC-III and (right) Arrhythmia datasets. The tight clustering around the identity line and narrow limits of agreement demonstrate high accuracy and minimal bias across both normal and pathological conditions.

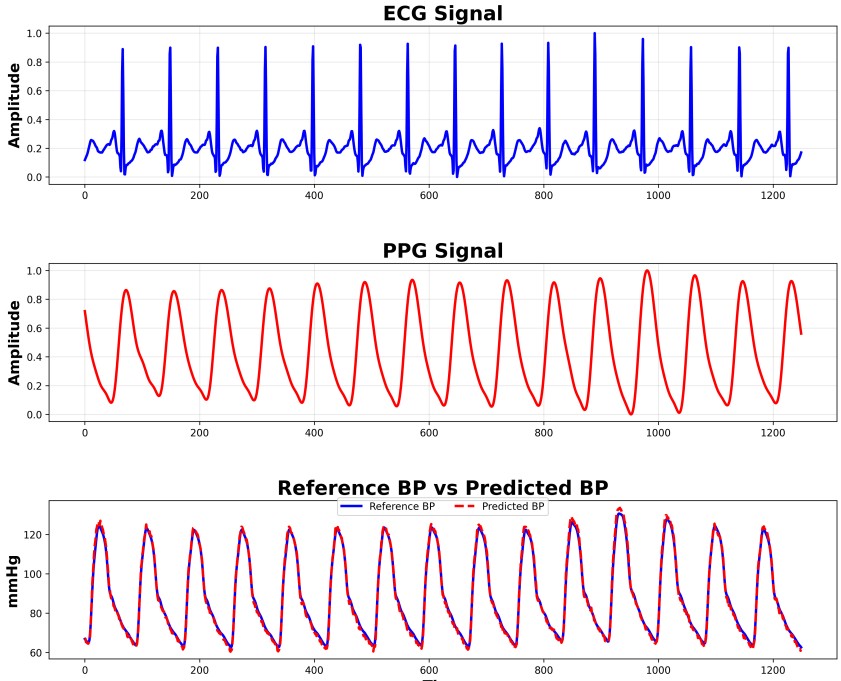

Figure 4: Representative examples of BP waveform generation by CausalDiffusion on MIMIC-III dataset. The generated waveforms (red) closely match the ground truth (blue) while preserving critical physiological features, including systolic peaks, dicrotic notches, and appropriate morphological adaptations to irregular rhythms.

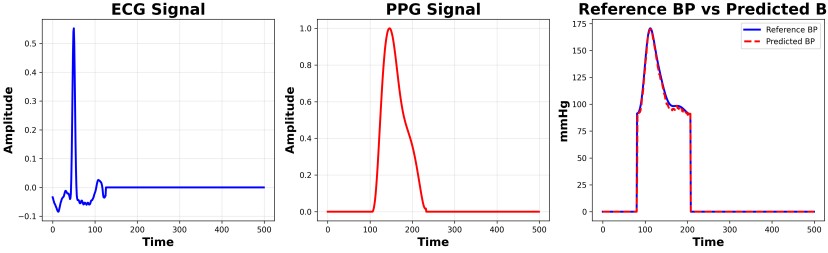

Figure 5: Representative examples of blood pressure waveforms generated by CausalDiffusion on the Arrhythmia dataset are presented. The results demonstrate excellent adaptability on individual arrhythmic samples.

# B    APPENDIX: PPG-TO-ECG SYNTHESIS DETAILS

This appendix provides supplementary details for the PPG-to-ECG synthesis task, including the features and components used to construct the prior causal graph, alongside additional visualizations of CausalDiffusion's performance.

## B.1    FEATURES, COMPONENTS, AND PRIOR CAUSAL GRAPH

The synthesis process maps a set of 15 physiologically meaningful features extracted from the input PPG signal (Table 4) to 6 key components of the target ECG waveform, which represent critical cardiac electrical characteristics (Table 5).

Table 4: PPG Features for ECG Synthesis.

| Domain | Feature | Description |
|---|---|---|
| Morphological | PPG_Peak_Amplitude | Maximum amplitude in cardiac cycle |
| | PPG_Valley_Amplitude | Minimum amplitude in cardiac cycle |
| | PPG_Rise_Time | Time from valley to peak |
| | PPG_Fall_Time | Time from peak to valley |
| | PPG_Pulse_Width | Duration of systolic upstroke |
| | PPG_Pulse_Interval | Time between consecutive peaks |
| Derivative | dPPG_Peak | Maximum first derivative |
| | dPPG_Valley | Minimum first derivative |
| | sdPPG_a | Second derivative a-wave (early systolic) |
| | sdPPG_b | Second derivative b-wave (late systolic) |
| | sdPPG_c | Second derivative c-wave (early diastolic) |
| | sdPPG_d | Second derivative d-wave (late diastolic) |
| Frequency | PPG_Freq_LF | Low frequency power (0.04-0.15 Hz) |
| | PPG_Freq_HF | High frequency power (0.15-0.4 Hz) |
| Variability | PPG_Variability | Coefficient of variation of pulse intervals |

Table 5: Target ECG Components.

| Component | Symbol | Normal Range | Clinical Significance |
|---|---|---|---|
| R-wave Amplitude | ECG_R_Amplitude | 0.5-3.0 mV | Ventricular depolarization strength |
| QRS Width | ECG_QRS_Width | 60-120 ms | Ventricular conduction velocity |
| RR Interval | ECG_RR_Interval | 600-1200 ms | Heart rate and rhythm |
| T-wave Amplitude | ECG_T_Amplitude | 0.1-0.8 mV | Ventricular repolarization |
| QT Interval | ECG_QT_Interval | 300-500 ms | Total ventricular electrical activity |
| Heart Rate | ECG_Heart_Rate | 50-120 bpm | Cardiac rhythm |

The prior causal graph for this task was constructed based on established cardiovascular physiological principles, connecting PPG morphological features to ECG amplitude components, temporal PPG features to ECG intervals, and frequency-domain PPG features to heart rate components.

## B.2    SUPPLEMENTARY VISUALIZATIONS OF ECG SYNTHESIS PERFORMANCE

The following figures provide further qualitative and quantitative insights into the performance of CausalDiffusion on the ECG synthesis task.

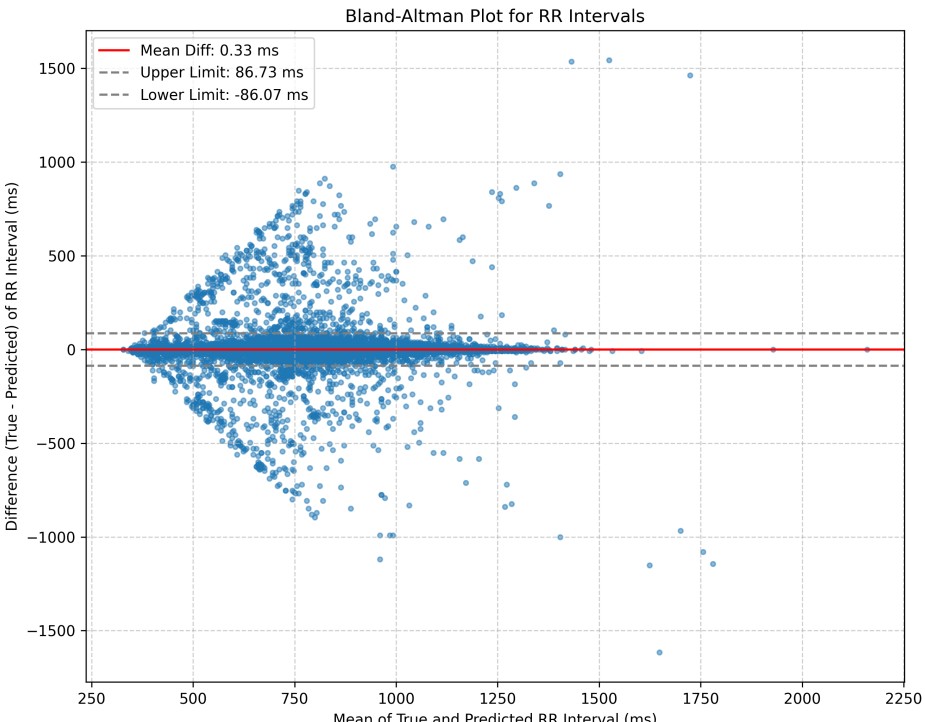

Figure 6: Bland-Altman plot for RR interval estimation on the test set. The plot assesses the agreement between the RR intervals derived from the ground truth ECG and those from the generated ECG. Each point represents a single RR interval, with the x-axis showing the mean of the true and predicted values, and the y-axis showing their difference. The solid red line indicates the mean difference (bias), which is extremely low at 0.33 ms. The dashed gray lines represent the 95% limits of agreement ($mean \pm 1.96 \times SD$), which are narrow at [-86.07 ms, 86.73 ms]. The plot demonstrates excellent agreement with no significant systematic bias, confirming the model's high accuracy in capturing heart rate dynamics.

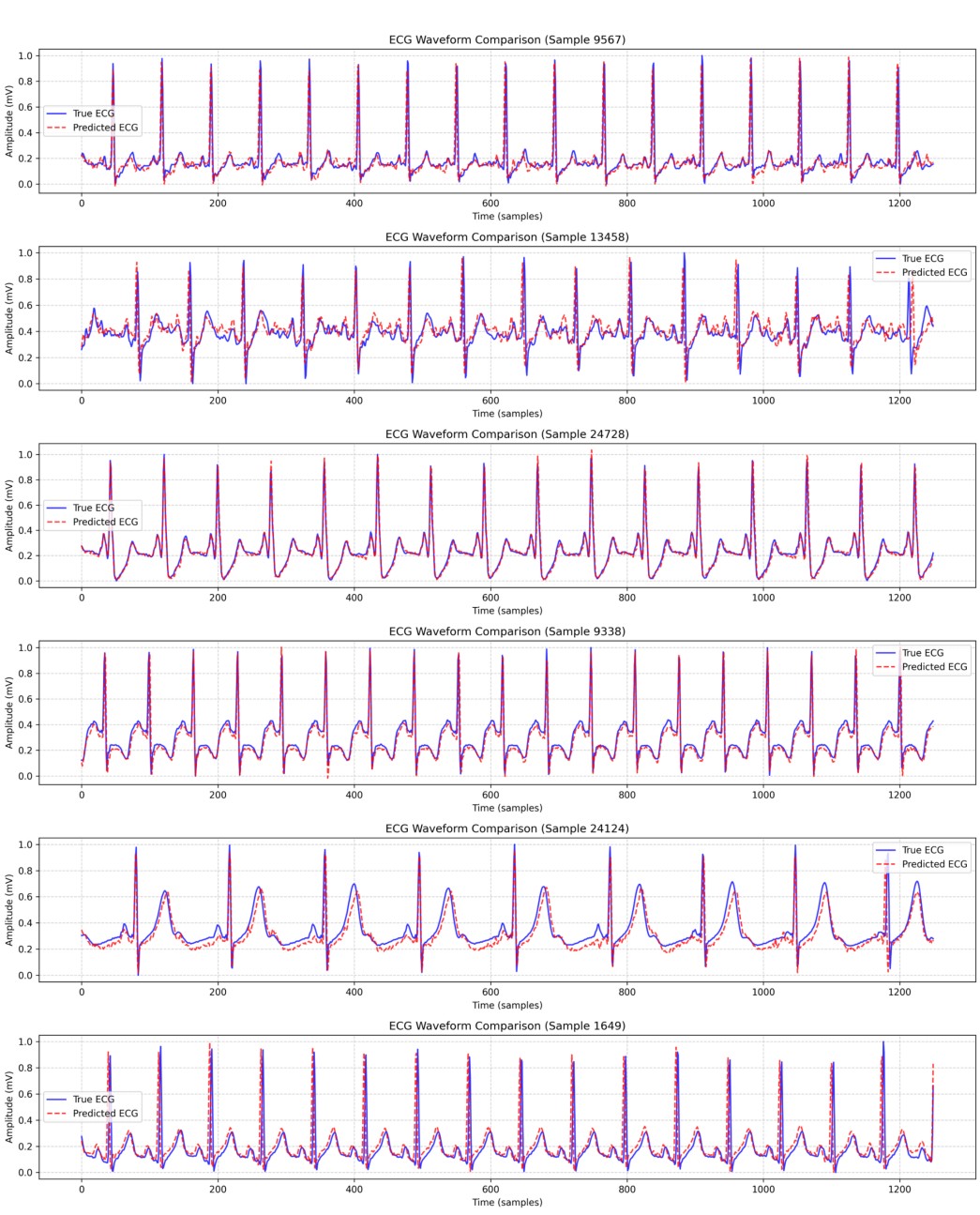

Figure 7: Qualitative results of PPG-to-ECG waveform synthesis on ten representative samples from the test set. In each subplot, the ground truth ECG waveform is shown in blue (solid line), while the ECG waveform generated by our CausalDiffusion model is shown in red (dashed line). The results demonstrate high morphological fidelity across a diverse range of subjects and signal patterns. Our model successfully reconstructs critical physiological features, including the P wave, QRS complex, and T wave, while maintaining a consistent and physiologically plausible rhythm.

