# OpenReview forum: "CausalDiffusion: A Causality-Embedded Diffusion Model for Cross-Modal Physiological Signal Synthesis"
_ICLR.cc/2026/Conference — Submitted to ICLR 2026_

### Official Review · Reviewer_XhC3 · 2025-10-24

**Soundness:** 3
**Presentation:** 3
**Contribution:** 3
**Rating:** 6
**Confidence:** 4

**Summary:**

This paper proposes using causal graphs to guide the stable and physiologically meaningful generation of signals. The experimental results are promising, and the research topic is interesting.

**Strengths:**

This paper proposes using causal graphs to guide the stable and physiologically meaningful generation of signals. The experimental results are promising, and the research topic is interesting.

**Weaknesses:**

(1) The experimental evaluation is comprehensive, as it jointly considers Clinical Parameter Accuracy and Morphological Metrics to assess the generated signals. However, I have a conceptual concern regarding the evaluation protocol. In real-world physiological processes, the mapping from PPG to ECG is inherently one-to-many due to underlying stochasticity and inter-individual variability. As noted in your model design, different noise inputs combined with the same PPG should yield diverse plausible ECG outputs. Therefore, directly computing RMSE and MAE between a single generated sample and the ground-truth signal may not fully reflect the generative nature of the task and may impose an unrealistic one-to-one correspondence. Would it be more appropriate to generate multiple ECG samples per input (by sampling different noise conditions) and report averaged RMSE and MAE values to better capture the probabilistic nature of the generation task?
(2) In Table 1, the units of RMSE and MAE under “Morphological Metrics” are shown as (mmHg). However, these metrics are meant to measure waveform reconstruction error rather than blood pressure values. Could the authors clarify whether this is a typographical mistake, or provide justification if these units are indeed intentional?
(3) Among the compared methods, I noticed that several are blood pressure estimation methods rather than data generation methods—e.g., CNN (Kokkhunthod et al. (2024)). What was the experimental setup for this comparison? This raises my concern: did the authors truly conduct a fair comparison? Specifically, how did the authors compare blood pressure waveform generation models (e.g., your CausalDiffusion) with blood pressure prediction models?
(4) While the integration of causal reasoning into diffusion models is an interesting direction, the current contribution appears somewhat incremental from an innovation standpoint. Specifically, the proposed framework seems to build upon an existing DDPM backbone by introducing causal guidance as an auxiliary module, rather than fundamentally rethinking the generative process.

I am generally positive about the paper’s contribution and appreciate the integration of causal modeling with diffusion-based signal generation. However, the concern raised in Comment (3) is substantial. If the authors can clarify the comparability of these baselines and justify the evaluation setup convincingly, I would be willing to increase my rating.

**Questions:**

(1) The experimental evaluation is comprehensive, as it jointly considers Clinical Parameter Accuracy and Morphological Metrics to assess the generated signals. However, I have a conceptual concern regarding the evaluation protocol. In real-world physiological processes, the mapping from PPG to ECG is inherently one-to-many due to underlying stochasticity and inter-individual variability. As noted in your model design, different noise inputs combined with the same PPG should yield diverse plausible ECG outputs. Therefore, directly computing RMSE and MAE between a single generated sample and the ground-truth signal may not fully reflect the generative nature of the task and may impose an unrealistic one-to-one correspondence. Would it be more appropriate to generate multiple ECG samples per input (by sampling different noise conditions) and report averaged RMSE and MAE values to better capture the probabilistic nature of the generation task?
(2) In Table 1, the units of RMSE and MAE under “Morphological Metrics” are shown as (mmHg). However, these metrics are meant to measure waveform reconstruction error rather than blood pressure values. Could the authors clarify whether this is a typographical mistake, or provide justification if these units are indeed intentional?
(3) Among the compared methods, I noticed that several are blood pressure estimation methods rather than data generation methods—e.g., CNN (Kokkhunthod et al. (2024)). What was the experimental setup for this comparison? This raises my concern: did the authors truly conduct a fair comparison? Specifically, how did the authors compare blood pressure waveform generation models (e.g., your CausalDiffusion) with blood pressure prediction models?
(4) While the integration of causal reasoning into diffusion models is an interesting direction, the current contribution appears somewhat incremental from an innovation standpoint. Specifically, the proposed framework seems to build upon an existing DDPM backbone by introducing causal guidance as an auxiliary module, rather than fundamentally rethinking the generative process.

---

### Official Review · Reviewer_5oCW · 2025-10-25

**Soundness:** 2
**Presentation:** 3
**Contribution:** 2
**Rating:** 2
**Confidence:** 3

**Summary:**

The paper proposes CausalDiffusion, a conditional denoising diffusion probabilistic model, DDPM, that embeds a sample specific causal graph into the reverse process for cross modal physiological signal synthesis. The static base graph is built offline by fusing domain knowledge with a Fast Causal Inference, FCI, graph, then modulated online to yield a dynamic adjacency used by a Graph Convolutional Network, GCN, whose embedding conditions the U Net denoiser. The method targets two tasks, ECG and PPG to ABP synthesis and PPG to ECG synthesis, reporting improved waveform fidelity and clinical parameter accuracy, including compliance with the AAMI blood pressure validation criteria. The paper provides equations for the forward and reverse processes, a multi objective loss with an auxiliary physiological component predictor, patient level splits, and comparisons against DDPM and several sequence baselines on MIMIC III and a proprietary arrhythmia cohort.

**Strengths:**

- Clear motivation, reducing physiologically implausible generations by injecting mechanistic structure into the reverse diffusion steps. The architectural decomposition into base graph construction, dynamic modulation, and conditioning is conceptually clean.
- Solid task choice for clinical relevance. The ABP task with ECG and PPG inputs and the robustness analysis on arrhythmia showcase a challenging, high value setting. Reported AAMI compliant SBP and DBP errors strengthen the practical angle.
- Ablation against a matched DDPM highlights the incremental value of causal conditioning, especially under arrhythmic patterns where out of distribution effects are stronger.
- The methodology is mostly mathematically consistent, with correct conditioning of the reverse transition mean and an explicit accounting of dynamic graph weights in the GCN aggregation.

**Weaknesses:**

- The claimed causal component rests on applying FCI to a static, hand engineered feature matrix pooled across samples. This discards temporal directionality and potential lag structure that are central in cardiovascular dynamics. State of the art time series causal discovery methods, such as PCMCI+, DYNOTEARS, and related approaches, directly model lagged dependencies and invariances under interventions or environment changes. The paper should either justify FCI on static features or adopt a method designed for time series with latent confounding. [1, 2, 3]
- The preprocessing step that removes one variable from any highly correlated pair before FCI, to guarantee invertibility for partial correlations, is a strong heuristic. It risks eliminating truly causal yet redundant variables and it introduces selection induced correlations. At minimum, sensitivity analyses across the correlation threshold and across alternative CI tests are needed.

- The training objective mixes L2 and L1 losses on the noise prediction without discussing scale balancing beyond a scalar λ. Since the auxiliary loss supervises a component predictor that subsequently defines the graph embedding, any mismatch in loss scales can bias the effective conditioning. Please report the variance of the noise term across timesteps, and provide calibration of $\lambda$ and $\lambda_{L1}$ via validation curves or an uncertainty analysis.
- The reverse process uses fixed variances, standard in DDPM, but physiological signals have heteroscedastic noise. Reporting learned variance experiments or at least a justification for fixed variance would strengthen the claim that improvements stem from causal conditioning rather than a favorable noise schedule, for example the cosine schedule.

- The ablation contrasts CausalDiffusion against a plain DDPM. Given the pipeline has several coupled pieces, please ablate: i) prior only, G_prior, ii) data driven only, G_data, iii) base graph Abase without dynamic modulation, iv) dynamic modulation with shuffled G_prior edges, v) removal of the auxiliary physiological predictor, and vi) replacement of GCN with a simple MLP. Without these, it is hard to attribute gains to causal guidance rather than to additional representation capacity.
- The dynamic modulation matrix M is central but opaque. Provide qualitative analyses of Adynamic distributions and case studies contrasting healthy versus arrhythmic samples to show that domain plausible dependencies, for example PTT related edges, are up weighted when appropriate.

- Cross dataset generalization is not evaluated. Both tasks are trained and tested within MIMIC III derived splits or a single site arrhythmia cohort. A credible causal prior should help under dataset shifts. Evaluate on an external cohort or at least perform leave patient group out evaluations stratified by demographics, sensor type, or acquisition site.
- Uncertainty and calibration are missing. For clinical usability, report prediction intervals for SBP and DBP and coverage under the AAMI envelope, as well as calibration curves for HRV metrics and QRS detection. Compare against conformal baselines.
- Some table entries appear inconsistent, for example Mamba on MIMIC III shows RMSE 5.16 and MAE 13.64, which violates MAE <= RMSE for non negative errors or suggests a unit mismatch. Please double check metric definitions and units, or explain if different normalizations were used.
- The proprietary arrhythmia dataset lacks details on labeling, patient selection, and signal pre processing decisions, for example zero padding to fixed length 500. Clarify whether padding artifacts influence evaluation windows and how beats are segmented. Provide inter rater or algorithmic quality control for arrhythmia annotations.

- The comparison set focuses on generic sequence models and DDPM. Recent causal generative modeling lines introduce explicit structural causal models, SCMs, or interventional controls within diffusion or likelihood based generators. The paper should position against these, even if tasks differ, and discuss whether CausalDiffusion supports counterfactual sampling or interventional queries. [4, 5, 6]
- Physics informed and constraint based training is a large, adjacent literature. The paper could compare against stronger priors like differentiable simulators or hybrid PINN diffusion models rather than only regularizers in the loss. [7, 8]

- Graph construction requires multiple hyperparameters, feature engineering choices, and statistical thresholds. A thorough appendix with exact feature lists per task, selection thresholds, CI test \alpha, and the learned modulation architecture would substantially aid reproducibility. Some of this exists, but several key items are currently high level.

References:

[1] Runge J. Discovering contemporaneous and lagged causal relations in autocorrelated nonlinear time series datasets. InConference on uncertainty in artificial intelligence 2020 Aug 27 (pp. 1388-1397). PLMR.

[2] Andrei Pamfil, et al. DYNOTEARS, Structure learning for dynamic causal models, AISTATS, PLMR 2020.

[3] Assaad CK, Devijver E, Gaussier E. Survey and evaluation of causal discovery methods for time series. Journal of Artificial Intelligence Research. 2022 Feb 28;73:767-819.

[4] Sanchez P, Tsaftaris SA. Diffusion Causal Models for Counterfactual Estimation. InConference on Causal Learning and Reasoning 2022 Jun 28 (pp. 647-668). PMLR.

[5] Mao C, Cha A, Gupta A, Wang H, Yang J, Vondrick C. Generative interventions for causal learning. InProceedings of the IEEE/CVF Conference on Computer Vision and Pattern Recognition 2021 (pp. 3947-3956).

[6] Xia KM, Pan Y, Bareinboim E. Neural Causal Models for Counterfactual Identification and Estimation. InThe Eleventh International Conference on Learning Representations.

[7] Raissi M, Perdikaris P, Karniadakis GE. Physics-informed neural networks: A deep learning framework for solving forward and inverse problems involving nonlinear partial differential equations. Journal of Computational physics. 2019 Feb 1;378:686-707.

[8] Deng Z, Tian H, Zheng X, Zeng DD. Deep causal learning: representation, discovery and inference. ACM Computing Surveys. 2025 Sep 8;58(2):1-36.

**Questions:**

1.  please justify using FCI on static features rather than a time series method with explicit lags. If retained, report sensitivity to the correlation threshold $\tau$, the CI significance level $\alpha$, and the choice of CI test. Can you show identical results when replacing FCI with PCMCI+ on lagged features, or explain observed differences?
 2.  please provide qualitative inspections of Adynamic across cardiac conditions, for example arrhythmia subtypes, and identify which edges change most. A small set of Sankey or heat map visualizations would help interpretability.
 3.  beyond the DDPM baseline, which component contributes most, G_prior, G_data, dynamic modulation, or the auxiliary predictor? A full ablation will help rule out capacity confounds.
 4. have you tried learned variance or noise conditioned U-Nets, and if so, are improvements maintained?
 5. can you report performance on a truly external dataset or a leave cohort out split that differs in device or demographic distribution, to assess whether the causal prior enhances robustness to dataset shift?
 6. please clarify the Mamba MAE anomaly on MIMIC III and any unit or normalization differences across methods.
 7.  for the proprietary arrhythmia cohort, please clarify consent, ethics approval, and any de identification or privacy preserving steps taken.

---

### Official Review · Reviewer_jge2 · 2025-10-30

**Soundness:** 2
**Presentation:** 2
**Contribution:** 2
**Rating:** 2
**Confidence:** 4

**Summary:**

This paper presents a method for cross-modal physiological signal (time-series) generation such that the generations are physiologically plausible and not just statistically consistent. It includes a causal vector G associated with causal graph (dynamic, sample-specific) as one of the conditional components in the noise prediction model (UNet) of the reverse diffusion process. This causal graph is  constructed based on a combination of  domain knowledge as well as discovered using  FCI algorithm  from training data. This static base is then modulated by a dynamic modulation matrix which generates the final causal vector that guides the U-Net denoiser. The evaluation of the method considered two public datasets for cross-modality signal synthesis considering several existing baselines.

**Strengths:**

The idea of including a causal graph to condition the denoising process is interesting and makes sense, especially for cross-modality signal synthesis.

The construction of the causal graph based on a combination of prior knowledge and causal discovery from data is also interesting.

**Weaknesses:**

This paper can be substantially improved in its clarity in the key methodological components, especially regarding the relation between the diffusion model and the causal graph. More specifically,
- It is not clear the are the “features” (that consists of V in the causal graph) that are extracted from the input signals — are these abstract or physiological signals, and whether the causality in time is modeled. This also raises questions about how is the data-driven causal graph discovered (see detailed questions below).
- “For each input sample, a set of initial node features is derived from its physiological characteristics” — It is not clear how this is done.
- Some concrete examples on the experimental datasets for the causal graph may be helpful to answer the above questions.


Similarly, clarifications are needed in the evaluation section, including
- Lack of clarifty about the waveform fidelity metrics and the mean error metrics (see detailed questions below)
- 4.2 was emphasized to be a cross-modal synthesis task, but ABP generation task (in 4.1) is not clear: isn’t it also cross-modality?

It is not clear why std is included in some results, but not others. In general, statistics from multiple seeds are needed in order to demonstrate any statistical significance of the margins of improvements presented, especially in Table 2 where the margins are small.

Since the generative model is based on causal graph, a discussion on whether/how it may be used for interventional/counterfactual generation would be helpful.

**Questions:**

What are the features V in relation to the L-dimensional signals z? Is each variable in V corresponds to one modality, or is the causal relation over time within a unimodal z-signal also modeled?

If the different v’s correspond to different modality of signals (e.g., ECG vs. PPG) without considering time, how is the FCI method applied to the signals over time?

Eq 12: When performing union of G_prior and G_datadriven, what if it leads to circular dependency? Is it kept as it is since it is a base/skeleton causal graph yet to be processed as sample-specific, or is one discarded to avoid cycle? If latter, which one is considered, that of G_prior or G_datadriven, and why? In Fig. 1, if we look at blue and orange nodes, the union would have created a cycle and it considered the path of G_prior.

The waveform fidelity metrics seem to be calculated between generated and GT waveforms: does that mean somehow there are some sort of pairing of training signals across modalities? How is the pairing done?

The metric of ME (mean error) is used for ABP synthesis: does this mean the error considers the sign of the error? If that’s the case, how would they be aggregated across samples? In Table 1,  ME is indicated to be the smaller the better, but why? Also it is not clear why the results from CausalDiffusion are bolded (to be the best?) as they are neither the lowest in magnitude or signed-form.

Please clarify on the task in 4.1

---

### Official Review · Reviewer_NzH5 · 2025-11-06

**Soundness:** 2
**Presentation:** 3
**Contribution:** 2
**Rating:** 2
**Confidence:** 4

**Summary:**

The paper proposes CausalDiffusion, a conditional diffusion model for cross-modal physiological signal synthesis (e.g., PPG to ECG). The central idea is to integrate a causal graph into the reverse denoising process. This graph is constructed from an offline base graph that merges domain-knowledge edges with edges discovered by Fast Causal Inference (FCI) algorithm on the training set, together with an online modulation step that assigns sample-specific edge weights via a learned matrix and a GCN. Overall, the general idea makes sense, and the application domain is practically important.

**Strengths:**

The discussion section illustrating how CausalDiffusion removes artifacts in ABP waveforms is interesting to me. The overall presentation is also clear and easy to follow.

**Weaknesses:**

I find the construction of the causal graph somewhat heuristic in several respects.

1. It is unclear why FCI was selected over many alternative causal discovery methods. Could this choice introduce algorithm-selection bias in the results? Moreover, how are the hyperparameters of FCI configured?

2. The prior graph is derived from correlation-based feature selection combined with domain rules, which may conflate association with causation?

3. The fusion strategy, using a hard union $E = E_{prior} \cup E_{data}$, also appears heuristic. How are potential conflicts in edge directions or inconsistencies between the two graphs resolved?

**Questions:**

1. Are there any hyperparameters in the Fast Causal Inference (FCI) algorithm? How are they selected or tuned in your experiments?

**Details Of Ethics Concerns:**

The authors included a non-anonymous GitHub link directly in the abstract, which clearly reveals the identity of at least one author (a student named Weixiu Qiu from the University of the Chinese Academy of Sciences). This violates the double-blind review requirement and constitutes a breach of submission integrity rules.

---

### Meta-Review · Area_Chair_AspL · 2025-12-29

**Summary:**

The provided code link belongs to a GitHub account with author identity revealed. Therefore this violates the anonymity rules for double-blind submission -- a desk reject is warranted.

**Reviewer Concerns:**

Reviewers voted for rejection.

No author rebuttal is provided.

**Reviewer Scores:**

Reviewers voted for rejection.

No author rebuttal is provided.

---

### Decision · Program_Chairs · 2026-01-26

Reject